# An Empirical Study of the Role of Higher Education in Building a Green Economy

**Wenjuan Gao [1], Xiaohao Ding [1,2,*], Ran Chen [1] and Weifang Min [1,2,3]**

[1]  Graduate School of Education, Peking University, Beijing 100871, China; gaowenjuan@pku.edu.cn (W.G.); chen_ran@pku.edu.cn (R.C.); wfmin@pku.edu.cn (W.M.)
[2]  The Institute of Educational Economics, Peking University, Beijing 100871, China
[3]  National School of Development, Peking University, Beijing 100871, China
*  Correspondence: xhding@pku.edu.cn

**Abstract:** The relationship between higher education and economic development has long been emphasized in the research on economics and education. Much of the existing literature focuses on the gross domestic product (GDP) as a core measure of a nation's economic accounting system, but this may neglect some negative effects of production, such as resource depletion and environmental damage. Under such circumstances, the concept of "green GDP" was conceived to consider environmental influence simultaneously with the economy. It is, however, only theoretically feasible due to the complexity in calculating environmental pollution and the unavailability of data about resource consumption. Considering the measurement problems, this paper proposes a new approach to indirectly estimate green GDP. Using this approach, we mainly explore the impact of higher education on economic growth, especially regarding the development of a green economy. Results show that (a) higher education plays a significant role in building a green economy, and (b) green GDP is more responsive to changes in higher education than the traditional GDP. This study provides empirical evidence for the substantial contribution that higher education makes in promoting green economic growth to achieve comprehensive sustainable development.

**Keywords:** higher education; green GDP; environment; sustainable development

## 1. Introduction

It has been widely recognized that education has a significant positive effect on economic growth. Theodore W. Schultz argued in the human capital theory that education can help accumulate people's human capital so as to enhance their productivity in labor market [1]; this has been confirmed by empirical research in different countries and regions [2,3]. Higher education, in this way, plays an even more prominent role in economic development. Not only does it cultivate high-quality labor to increase the productivity of the whole society, but it also promotes technological and institutional innovation in order to improve the efficiency in production. Moreover, it should also be noted that higher education benefits people with knowledge and skills, and also changes their daily behaviors, or even shapes people's views and values in every way. All these possible effects of higher education on people will ultimately exert some impact on economic development, since the labor force is one of the basic factors in production [4]. Therefore, it is quite necessary to comprehensively explore the role of higher education in economic growth from different perspectives.

Previous research on higher education and economic development mostly employed traditional GDP as the measure of economic growth; however, this practice has long been critiqued, with the concern that GDP cannot accurately reflect the welfare of a nation. Even Richard Stone, one of the creators of the original GDP indicator, suggested that "the three pillars on which an analysis of

society ought to rest are studies of economics, socio-demographic and environmental phenomena" [5]. The concept of green GDP emerged in this context, aiming to make up for the shortcomings of traditional GDP accounting. In contrast with the limitations of traditional GDP, green GDP essentially stands for the net positive effect of national economic growth. Nevertheless, there were many difficulties in calculating green GDP in empirical studies. Intuitively, we can get green GDP by deducting the costs of environmental consumption and pollution from the traditional GDP, but this is mainly theoretically feasible considering the complexity in calculating environmental pollution and the unavailability of data about resource consumption. Given these research gaps, is it possible to propose a new approach to indirectly estimate green GDP? How does higher education affect green economic development? Is there any difference of the role higher education plays in green GDP and in the traditional GDP? Our paper contributes by answering these three research questions. The contribution and limitations of prior literature in this field need to be reviewed and analyzed before we continue our research.

## 2. Literature Review

This section mainly consists of three parts: First, it brings in the typical Solow–Swan growth model as well as its modified versions that are commonly used considering economic growth, which will also prepare our analysis later in the research design section; second, it presents a review of previous studies about the impact of higher education upon economic development, based on which the third part identifies and elaborates the research gaps, i.e., the limitations of traditional GDP measure and the difficulties in calculating green GDP in empirical research.

### 2.1. Economic Growth Model

Solow–Swan growth model is one of the classical models of economic development, which is also known as neoclassical economic growth model or exogenous economic growth model. It was developed by Solow to theoretically analyze the relationship between savings, capital accumulation, and economic growth within the framework of neoclassical economics [6]. The basic form of Solow–Swan model is as follows:

$$Y = AK^{\alpha}L^{1-\alpha}. \tag{1}$$

Y refers to the economic growth measured by GDP, while the three independent variables $K$, $L$, and A, respectively, represent capital, labor, and other factors in production such as technological development. This fundamental model has been developed in later research, mostly by separating other influencing factors from the variable A. For example, Barro and Sala-i-Martin [7] modified this model by taking knowledge and technology (variable T) into account:

$$Y(t) = F[K(t), L(t), T(t)]. \tag{2}$$

Mankiw, Romerm and Weil [3] developed a Mankiw–Romer–Weil version of the model considering human capital (variable H) in their analysis:

$$Y(t) = F[K(t), L(t), H(t)]. \tag{3}$$

Furthermore, the "new growth theory" developed by Romer and Lucas highlighted the role of knowledge and technology and suggested technological progress was the determining factor to ensure sustainable economic development.

These modifications of Solow–Swan model may be used in various contexts based on their applicability. The Mankiw version, for instance, can be used to explain why capital always flows from labor-intensive poor countries to the developed countries. In general, the Solow model has been recognized as the most fundamental and commonly used version in pertinent studies of economic growth, which will also be employed in our study.

## 2.2. Impact of Higher Education on Economic Growth

Prior research about the impact of education on GDP were often based on human capital theory, according to which human capital is tightly related to education and makes great contribution to economic growth. Lucas proposed that human capital is accumulated by both education in school and learning in practice. School education forms general human capital, which determines, to a large extent, the accumulation of specialized human capital generated via work experience. In workplaces, it is difficult to acquire human capital for people with a low educational level [8]. Therefore, education, serving as a prerequisite for human capital accumulation, is commonly employed as a proxy variable when analyzing the significance of human capital on economic development.

Most of the empirical research consistently concluded the positive effects of education at all levels on economic growth. Kyriacou proved that the stock of human capital was positively related to a nation's economic growth by using the data on average years of schooling as a proxy for human capital and regressing the Lucas endogenous economic growth model [9]. Barro and Lee explored how a set of quantifiable explanatory variables gave rise to differences in growth rates across countries, identifying the significant role of secondary-school attainment in the growth regression [2]. Apart from the general consideration, there were some research looking into the role of education in different contexts. For example, Yang, Gong, and Zhang used the panel data of 29 provinces in China during the period 1985–2000 and analyzed how education influenced economic development with the Solow model, measuring education by the proportion of people at least graduating from junior high school among the population aged 15 and over. It turned out that education had a significant positive impact on economic growth and the coefficient was much higher compared to other similar international empirical studies [10]. Interestingly, Wang, Fan, and Liu measured the educational level with an index calculated from average education years of the working population, and the coefficient in the Solow model indicated much less of an effect of education on economy in China in comparison with other countries like the United States [11]. Notwithstanding the disaccord in different degrees of the importance of education by using different measures, they did agree on the key role of education in promoting economic growth. In addition, there were also some studies comparing the heterogenous effects of different educational levels on economic growth, highlighting that globally, countries with higher enrollment rates in secondary and higher education have grown faster economically [12].

Particularly for the impact from higher education, Yue made an international comparison of the status of higher education and economic development from 1978 to 2017, and the results indicated that there was heterogeneity among different countries, with a more pronounced influence in developed countries [13]. Gyimah-Brempong, Paddison, and Mitiku found a significant relationship between higher education capital and the growth rate of per capita income in African countries [14]. Tin-Chun Lin focused on the role of higher education on economy using data from 1965 to 2000 in Taiwan, and found that higher education, particularly the fields of engineering and natural sciences, contributed greatly to economic growth [15]. Song and Wang showed that the labor productivity of higher education graduates was 2.17 times higher than that of primary education graduates; yet the contribution to economy was limited to the small scale of higher education in China [16]. Despite different empirical conclusions of relevant studies with various databases and methodologies, a general consensus has been reached that higher education promotes the sustainable economic growth and human capital may be the core driving force for sustainability.

## 2.3. Research Gaps

### 2.3.1. Limitations of Traditional GDP as a Measure of Economic Growth

In the existing literature, the traditional national economic accounting system has long been used to estimate the contribution of education to economic growth. Dating back to the 1940s, many Western countries pursued Keynesianism, which inspired government intervention in the national economy. In light of the government involvement, it was essential to analyze economic

development macroscopically. To achieve this goal, Kuznets, Epstein, and Jenks [17] then proposed the concept of gross national product (GNP), from which the gross domestic product (GDP) derived. Later, the United Nations adopted GDP as an important indicator of economic growth worldwide. At that time, the theory of property rights still needed to be further developed and improved, while natural resources and the ecological environment were regarded as free public goods, so they were excluded from the accounting system.

Nevertheless, in recent years, environmental deterioration with the shortage of resources globally has posed an unprecedented and grave threat to human development. This has raised people's awareness of the need for environmental preservation. Under this circumstance, the limitations of traditional GDP became evident. On the one hand, human economic activities have positively influenced society by creating wealth; on the other hand, the same activities have brought in negative effects by hindering the development of social productivity in many forms. For instance, relentless overexploitation has resulted in the diminishing supply of natural resources. Moreover, the discharge of sewage and waste, as well as deforestation, have been major contributors to environmental degradation. These downsides, however, have not received adequate attention. The current national economic accounting system only looks at the bright side of economic activities, which does not reflect practices of sustainable development.

Taking the trajectory of the Chinese economy as an example, in the year 1980, China was the most populous yet one of the poorest countries. Within approximately three decades, however, it has taken a significant leap, becoming the world's second largest economy only after the United States, which was exclaimed by many international media as the "Chinese miracle" [18]. Unexpectedly, in recent decades, the Chinese economic boom has started to tail off, with some social conflicts—previously concealed by the economic prosperity—standing out. One of the major problems was the imbalance in national industrial structure. The central government was forced to cut capacity in sectors such as coal and steel and to facilitate the deleveraging process, which, inevitably led to mass unemployment [19]. Furthermore, China's development has overlooked the destruction of environment and immoderate consumption of non-renewable resources; this, in turn, brought about environmental deterioration, such as more serious water pollution and an increase in carbon dioxide emission. Fortunately, this issue has been receiving more attention than before, given the improved living standards of Chinese people and increased public awareness in environmental protection. With this particular case of China, it is apparent that the traditional measurement of economic development merely by GDP may be misleading, and we are in urgent need to embrace new approaches to gauge economic growth, thereby guaranteeing the long-term benefits for human beings.

2.3.2. Difficulties in Measuring Green GDP in Empirical Analysis

"Green GDP" was conceptualized to consider environment on an equal footing with economy. It refers to the results of economic activities while also considering their impacts on natural resources (mainly including land, forests, minerals, water, and oceans) and the environment as a whole (such as ecological, natural, and human environments) [20]. In other words, green GDP takes into account the costs of resource depletion and environmental degradation incurred in economic activities. Green GDP is regarded as the indicator for sustainable economic development for several reasons. First, it measures the actual achievements in productivity in order to avoid pure pursuit of economic growth rate that neglects the externalities of economic activities. Second, it mirrors the scenarios of social welfare and progress, highlighting the importance of coordinating harmonious development of man and nature. Meanwhile, green GDP helps enhance public awareness in environmental protection and promote the transformation of development patterns. Nevertheless, this does not mean that the traditional GDP should be replaced by green GDP as a better solution. The traditional GDP is still the most important and direct indicator reflecting the levels of national economic development, while green GDP serves as a supplement in an ecological manner.

As early as 1971, the concept of "Eco-Requirement Indication (ERI)" was proposed by the Massachusetts Institute of Technology to reflect the relationship between economic growth and environmental resource pressure [21]. In the 1980s and 1990s, the World Bank tried to spread "green accounting" [22] and established the system of environmental economic account (SEEA) in some countries. However, this approach has not been applied extensively and most countries and regions barely take into account their natural resources and environmental conditions nationwide when assessing the economic growth. For example, China, in 2006, first published the research report of green national economic accounting of 2004, but there were no subsequent reports due to the accounting difficulties and data unavailability.

Generally, most of the previous research on green GDP were still at the exploratory stage, trying to discuss and develop green GDP theoretically considering its calculation difficulties. Boyd discussed the possibility to measure non-market value of natural resources in his article and proposed that national culture and social stability should be included when evaluating green GDP [23]. Li and Fang employed the ecological and geological methods to measure the consumption of global natural resources, with the purpose of calculating the green GDP of different countries [24]. There were also some other scholars who made intensified efforts to calculate green GDP using the input–output model. This model assessed the national inputs and outputs in industry, energy, transportation, and agriculture based on the environmental and economic account of World Bank. On this basis, the scholars adjusted GDP values by considering the actual consumption in various sectors. So far, they have calculated green GDP for different countries and regions, including Australia [25], Austria [26], Brazil [27], and Italy [28,29], the Netherlands [30], Sweden [31], the United Kingdom [32], and the United States [33], etc. Even so, it was still not easy to measure the resources and environmental inputs required in various industries, so this input–output model posed similar challenges in complexity and precision during calculation compared to the previous models. Furthermore, other studies focused on the determinants of green GDP. Talberth and Bohara found that the openness of countries was significantly negatively correlated to national green GDP, while there was a positive correlation between the countries' openness and the difference between their green GDP and traditional GDP [34].

Overall, higher education has positive effects on economic growth. Nevertheless, when measuring economic growth, traditional GDP may neglect some negative effects of production, such as resource depletion and environmental damage; while green GDP, though considering environmental influence simultaneously with the economy, is only theoretically feasible due to the complexity in calculating environmental pollution and the unavailability of data about resource consumption. Our paper aims to fill these gaps by proposing a new approach to indirectly estimate green GDP, and identifying how higher education exerts differential effects on green GDP and on the traditional GDP.

## 3. Research Design

### 3.1. Hypotheses

Our paper proposes a new approach to indirectly estimate green GDP, and based on this indicator, it is designed to address two main questions: (1) How does higher education affect green economic development? (2) Is there any difference of the role higher education plays in green GDP and in the traditional GDP? According to the prior literature, the traditional GDP is inextricably linked with higher education, and it is quite meaningful to evaluate the effect of higher education on green economic growth. Furthermore, to compare the different impacts on green GDP and the traditional GDP will offer us an in-depth understanding of the role of higher education. It is very convincing that countries with higher levels of education on average may have higher quality of human capital and more optimized industrial structures, which allow them to utilize resources in a more efficient and environmentally friendly way. Thus, this paper puts forward two main hypotheses to verify.

**Hypothesis H1:** *Higher education has positive influence on building green economies.*

**Hypothesis H2:** *Green GDP is more responsive to changes in higher education than the traditional GDP.*

*3.2. Green GDP Calculation*

Though the concept of green GDP has been proposed long ago, prior research had not made satisfactory achievements, given that there was no sufficient data to support the theoretical models by deducting costs of natural resources and environmental pollution from the traditional GDP. Our paper tries to measure green GDP from the perspective of the efficiency of energy resources utilization, which is regarded as one of the fundamental differences between the traditional and green economic development. In practice, we adopt two variables accordingly, one of which is the rate of energy use, calculating how much monetary output can be produced by one unit of energy resources. The other refers to the ratio of renewable energy of the total, which is more about assessing the damages caused by production. Therefore, we create a new indicator "green GDP" here to assess the development of green economy to some extent:

$$\text{GreenGDP} = \text{GDP} * energy * renew. \tag{4}$$

The variable *energy* is measured by the GDP produced through consuming per unit of energy resources, while *renew* represents the proportion of renewable energy of the total, an indicator of environmental damages. It can reflect the environmental pollutions caused by excessive consumption of fossil fuels and their irreparable dangers and damage to human society. Usually, the higher the share of renewable energy is, the more clean energy can be used and the less waste will be produced. Here, the term "green GDP" stands for both the domestic productivity and energy efficiency as well as environmental preservation of a country or region.

In this way, the rates of energy utilization and the proportion of renewable resources are much more accessible. However, this method has obvious weaknesses. It cannot be considered as the real green output of a country, but merely an indicator logically related to green GDP, so the absolute value generated from this equation has little practical significance. In spite of this, the indicator still reveals the costs in resources and environment to a certain extent. When studying the economic growth of countries, usually we are more concerned about the relative changes than the absolute values of economic output. Therefore, it is still meaningful to focus on the diachronic changes and cross-country comparison by applying this approach in economic analysis.

*3.3. Modified Solow–Swan Model with Higher Education*

In order to identify the influence of higher education on green GDP, this paper first incorporates variables of higher education into the Solow model. Based on the human capital theory, economic development depends not only on the quantity of labor, but also on their quality. In this way, it is not enough to just include the number of human capitals, namely the variable *L* in the Solow model. The quality of labor should also be taken into account, which can be reflected by the overall educational level of a country.

The newly proposed form can be written as follows:

$$\text{Y} = \text{A}K^{\alpha}L^{\beta}E^{\gamma}. \tag{5}$$

Y here refers to either the traditional GDP, or the green GDP, and the independent variables *K*, *L*, and A respectively stand for capital, labor, and other factors in production. The variable *E* represents the quality of labor and its proxy variable in our research is the gross enrollment rate of higher education.

*3.4. Modelling the Impact of Higher Education on Green GDP*

The empirical analysis of our paper mainly consists of two parts. The goal of the first stage is to verify the first hypothesis mentioned above, i.e., whether higher education has positive influence on building green economies. We standardize the traditional and green GDP of countries and regions in our sample using the following equations.

$$\text{standardized GDP} = \frac{\log(GDP) - \overline{\log(GDP)}}{SD(\log(GDP))}, \tag{6}$$

$$\text{standardized green GDP} = \frac{\log(green\ GDP) - \overline{\log(green\ GDP)}}{SD(\log(green\ GDP))}. \tag{7}$$

The standardized GDP and green GDP are both lognormal distributed with a mean of 0 and variance of 1. The values of normalized GDP and green GDP represent the position of the original values in the overall distribution. In this way, the differences between the standardized green and traditional GDP can be a signal to determine whether the national economy is greener.

$$GAP = \text{standardized green GDP} - \text{standardized GDP} \tag{8}$$

If the difference, or the variable *GAP*, is positive, then the country's green GDP ranks higher in the overall distribution than the conventional GDP, indicating its national economy is green; otherwise, if *GAP* is negative, the nation's GDP outranks its green GDP, which means the corresponding economic growth is not so sustainable and environmentally friendly. Here, we build a general linear regression of *GAP* on higher education.

$$GAP = \pi_0 + \pi_1 * K + \pi_2 * L + \pi_3 * E \tag{9}$$

In the above equation, *GAP* serves as the dependent variable. The independent variables *K* and *L* represents capital and labor separately; while *E* stands for the gross enrollment rate of higher education, whose coefficient reveals the effect of higher education in the national green economic development.

The second part compares the contribution rates of higher education to the traditional and green GDP growth. By taking the logarithm on both sides of Equation (5), we build log–log models of green GDP as well as the traditional GDP.

$$\ln(\text{GDP}) = \ln(A) + \alpha_1 \ln(K) + \beta_1 \ln(L) + \gamma_1 \ln(E) \tag{10}$$

$$\ln(\text{Green GDP}) = \ln(A) + \alpha_2 \ln(K) + \beta_2 \ln(L) + \gamma_2 \ln(E) \tag{11}$$

After regressing the traditional GDP and green GDP separately on capital, labor and higher education, we plan to compare the parameters of these two models in order to explore differential impacts of factors pertaining to production on the green and traditional GDP. It should be noted that the output of these two models are both elasticity coefficients, measuring the percentage change in the traditional GDP and in the green GDP as a result of a percentage change in capital per capita, labor per capita, and enrollment rate of higher education, respectively. Given that elasticity is a dimensionless measure of the sensitivity or responsiveness of one variable to changes in another, the coefficients in the two models are easily interpreted and compared across categories.

## 4. Data and Empirical Results

*4.1. Data Description*

The data source of our study is mainly from the World Bank, and we particularly select variables including GDP per capita (constant 2010 US $) of different countries and regions, their total population,



total capital (constant 2010 US $), labor population (total number), total enrollment rate of higher education, GDP generated by per unit energy consumption (constant 2011 PPP US $ per kg oil equivalent), and ratio of renewable energy.

We mainly choose relevant data during the period 1990–2015. This is driven by two reasons, the first is the United Nations Development Program had not started to collect data about the average years of schooling worldwide before the year 1990, and the second is that the statistics after the year 2015 have not been updated as of April 2019. Furthermore, we kept 187 countries and regions, which are among the 189 member states of the World Bank and also among the 193 member states in the United Nations. We excluded from our sample countries that deviate from general patterns of economic development, such as Kashmir and some post-war Middle Eastern countries. By doing this, we ensured our sample was representative, thereby improving the quality of our dataset in order to get more accurate results.

## 4.2. Comparison of Green GDP across Countries

Green GDP comes from and is tightly related to the traditional GDP; however, it serves as an irreplaceable measure for overall economic development different from the traditional indicator, because it takes into account the energy and environmental costs. As indicated in Table 1, we found that in 2014, there were wide gaps for different countries and regions after comparing their standardized GDP per capita with the standardized green GDP per capita. We chose the year 2014 because the data then were the most complete. It is evident that countries with higher green GDP per capita usually have highly developed tertiary industries, which consume few natural resources and do not cause much damage to the environment. For example, in Switzerland, whose green GDP ranked much higher than its traditional GDP (i.e., positive *GAP*), the financial industries, as well as other services, contribute to the vast majority of its GDP, while heavy industries and agriculture account for only a negligible share. By contrast, countries with negative *GAP* are often characterized by imbalanced industrial structures. For instance, the economic growth for nations in the former Soviet Union relies heavily on the energy-consuming and polluting industries, while those oil exporting countries like Iran are dependent on national natural resources. In fact, after comparing the *GAP* of the countries and regions, we learn that countries with outstanding performance in green GDP are not necessarily highly developed in economy and technology. Many newly emerging countries perform quite well with moderate economy scale and stable social development. In this way, whether a nation's economy is green is not merely determined by its power in economics, science, and technology, but relies more on its development pattern. For some nations that achieved industrialization earlier, economic stagnation refrained them from timely industrial restructuring and technological innovation, which, inevitably, set back their green economy development. On the other hand, the latecomers may have learned the lessons and benefited from current technological advances.

**Table 1.** Comparison of *GAP* across countries and regions.

| | Standardized GDP Per Capita | Standardized Green GDP Per Capita | *GAP* (Per Capita) |
|---|---|---|---|
| Iran | 0.11 | −1.42 | −1.53 |
| Iraq | 0.02 | −1.17 | −1.20 |
| United Arab Emirates | 1.39 | −1.20 | −2.59 |
| Saudi Arabia | 0.98 | −3.17 | −4.15 |
| Ukraine | −0.32 | −1.32 | −1.00 |
| Belarus | 0.15 | −0.33 | −0.48 |
| Russia | 0.56 | −0.47 | −1.03 |
| United States | 1.57 | 0.94 | −0.63 |
| Germany | 1.49 | 1.31 | −0.18 |
| Japan | 1.51 | 0.86 | −0.65 |
| Switzerland | 1.84 | 2.10 | 0.25 |
| Denmark | 1.68 | 2.02 | 0.34 |
| Sweden | 1.61 | 1.93 | 0.32 |
| Norway | 1.95 | 2.38 | 0.43 |
| China | 0.13 | −0.09 | −0.22 |

**Note**: *GAP* is an indicator to determine whether a nation's economy is green; *GAP* (per capita) equals standardized green GDP per capita minus standardized GDP per capita.

### 4.3. The Role of Higher Education to Green GDP

In order to examine the role of higher education, we performed the two-step analysis elaborated in the research design section. At the first stage, we ran the linear regression of Equation (9) based on the per capita data to verify the first hypothesis that higher education has positive influence on building green economies. The dependent variable in the model was *GAP* (the difference between standardized green GDP per capita and standardized GDP per capita), while the explanatory variables were capital per capita, labor per capita, and the gross enrollment rate of higher education. After checking with the Hausman test, we used the country fixed effect model with the panel data, and the regression results are shown in Table 2. The coefficient $\pi_3$ demonstrates the relationship between *GAP* and the gross enrollment rate of higher education. It is apparent that the enrollment rate of higher education had a statistically significant positive influence on the outcome *GAP*, indicating our first hypothesis to be valid.

**Table 2.** The linear regression of *GAP* on higher education.

| | *GAP* |
|---|---|
| $\pi_1$ (Capital per capita) | $9.11 \times 10^{-6}$ * |
| | $(4.38 \times 10^{-6})$ |
| $\pi_2$ (Labor per capita) | −0.40 * |
| | (0.24) |
| $\pi_3$ (Gross enrollment rate of higher education) | $5.26 \times 10^{-3}$ *** |
| | $(4.36 \times 10^{-4})$ |
| _cons | -0.05 |
| | (0.10) |
| N | 2078 |
| group | 128 |

**Note:** t statistics in parentheses; * $p < 0.1$, ** $p < 0.05$, *** $p < 0.01$.

As for the second hypothesis that green GDP is more responsive to changes in higher education than the traditional GDP, we ran log–log regressions of Equations (10) and (11) separately at the per capita level. The variable higher education was still calculated by the gross enrollment rate of higher education. According to the regression results, a percentage increase in the enrollment rate of higher education can significantly lead to 0.2% of growth in GDP per capita, while green GDP per capita can significantly rise by 0.33% with one percentage increase in the enrollment rate of higher education.

Furthermore, the comparison of estimations from the two regression models (suest), combined with the chi-square tests, show that the coefficient of education in the green GDP model was significantly larger than that in the GDP model at the per capita level (Table 3). Considering that the regressions were modified from the Solow growth model, the coefficients can be interpreted in the same way. Therefore, we can conclude that green GDP is more sensitive to changes in higher education than the traditional GDP, which verifies the second hypothesis.

**Table 3.** The log–log regressions of gross domestic product (GDP) and Green GDP on higher education.

|  | GDP Per Capita | Green GDP Per Capita | SUEST |
|---|---|---|---|
| $\alpha$ (Capital per capita) | 0.23 *** | 0.42 *** | chi2 = 24.62 |
|  | (0.02) | (0.04) | Prob>chi2 = 0.0000 |
| $\beta$ (Labor per capita) | 0.86 *** | 1.66 *** | chi2 = 9.20 |
|  | (0.08) | (0.29) | Prob>chi2 = 0.0024 |
| $\gamma$ (Gross enrollment rate of higher education) | 0.20 *** | 0.33 *** | chi2 = 9.95 |
|  | (0.01) | (0.02) | Prob>chi2 = 0.0016 |
| _cons | 6.72 *** | 11.36 *** |  |
|  | (0.08) | (0.19) |  |
| N | 2113 | 2113 |  |
| group | 131 | 131 |  |

**Note:** t statistics in parentheses; * $p < 0.1$, ** $p < 0.05$, *** $p < 0.01$.

## 5. Discussion

### 5.1. Possible Mechanisms of Higher Education on Green Economic Development

This paper evidenced that higher education plays a significant role in developing a green economy. In order to better understand these underlying mechanisms, Figure 1 summarizes several possible channels of the corresponding impact. First of all, from the perspective of economic development itself, the human capital theory has brought education to the attention of economists and it has been unanimously recognized by academia that education can improve the total factor productivity and promote economic growth by enhancing the general quality of human capital. Institutions of higher education, undoubtedly, have been considered as cradles for top talents, which help to improve the labor quality by providing human capital to the economy.

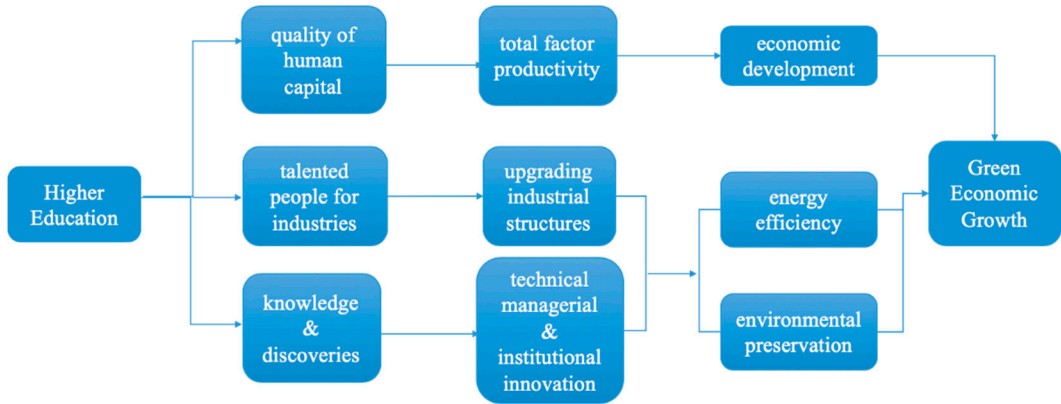

**Figure 1.** Mechanism of how higher education influence green GDP.

Apart from forming human capital, higher education also promotes economic development by accelerating industrial optimization and upgrading industries from labor-intensive to capital-intensive, and finally to knowledge- and technology-intensive [35]. A timely optimization of industrial structures

is of great significance not only for economic progress, but also to ensure a smooth and sustainable operation of the green economy, during which nations can reduce their consumption of energy resources and control the costs in the environment. Indeed, among the many factors affecting industrial upgrading, well-educated workers with sufficient knowledge and skills who can adapt well to highly advanced technology-intensive industries make a fundamental and critical difference. Higher education, by engaging talented people who are in pursuit of knowledge and skills, is an essential prerequisite for industrial upgrading. In this way, higher education plays a leading role in the optimization of industrial structures so as to achieve sustainable economic growth.

Another possible channel is that higher education inspires individuals to acquire knowledge and open their minds in order to pioneer new discoveries and bring about technological, managerial and institutional innovations. These innovations lead to progressive methods of production for all walks of life and drive new demands for a larger size of the economy. Furthermore, these innovations streamline production processes to improve the efficiency of the green economy [36].

In this way, through three distinct channels—high-quality human capital, talented people for industries, and innovations in knowledge and technology—higher education is critical for green economic growth.

### 5.2. Implications for Sustainable Development: Taking China as An Example

Higher education not only contributes greatly to expanding the size of economies, but more importantly, it helps to promote economic growth in a more environmentally friendly way. This is particularly meaningful for countries facing challenges in sustainable development. The context in China, whose development path was found problematic and not sustainable, has been discussed in our research as a typical example.

Over the past four decades since the reform and opening up, China has maintained a high economic growth rate, which can be attributed to either the institutional reform or the large population. However, it has been in a precarious situation considering the serious environmental pollution, waste of resources, and the aging of population, etc. Fortunately, the Chinese economy has started to pay attention to the sustainability of its economic growth, not just striving for higher productivity. The society now stipulates that people should seek comprehensive development instead of simple growth in GDP, aiming at sustainable growth in a socialist market economy. With this principle, its development path has been adjusted from three dimensions, the first and most obvious of which is to slow down the growth rate and develop at a modest pace. At the same time, it emphasizes continuous optimization and upgrading of industrial structures. Another noticeable characteristic is to shift the driving force from factor- and investment-driven to innovation-driven. Data of China's first three quarters in 2018 depicted a better economy with higher quality and more optimized structures. Specifically, consumption contributed more to economic growth than investment, service sector grew faster than the secondary sector with high-tech industries and equipment manufacturing standing out, and energy consumed by per unit of GDP declined noticeably. China's tertiary sector has contributed substantially to the economic growth, accounting for more than half of the total industries (Figure 2), and its proportion is predicted to increase steadily in the future. Overall, China has grown from a large agricultural country to a world factory, after which it has developed as a technologically innovative economy by creating wealth through science, technology, and culture.

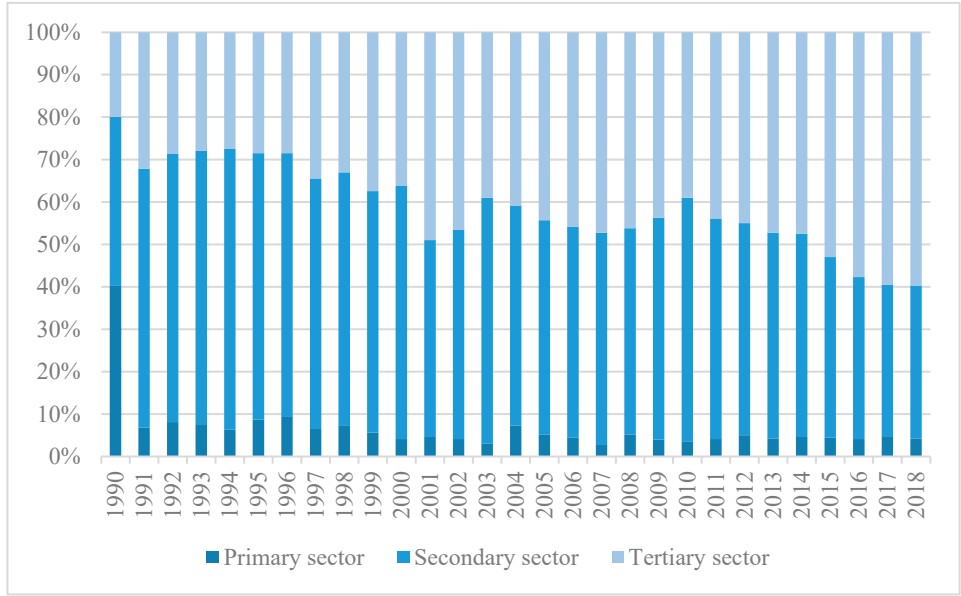

**Figure 2.** Changes of Chinese industrial structures.

As for the educational investment in China, before the year 2000, its educational expenditures were less than 2% of GDP, which was much lower than the average level worldwide, not to mention in comparison with the spending in developed countries. However, China's current investment in education has increased to more than 4% for six consecutive years, of which expenditures in higher education constituted about 1.4%, equivalent to most developed countries. This, with no doubt, is a positive signal for China's future development. By transforming the advantage of a large population in China into the feature of high-quality talents, higher education will definitely play a crucial role in China's sustainable development in the future.

## 6. Conclusions

This paper explores the impact of higher education on the development of a green economy. By proposing a new approach to indirectly estimate green GDP, two hypotheses have been confirmed through empirical analysis.

First, higher education plays a significant role in building a green economy.

Second, green GDP is more responsive to changes in higher education than the traditional GDP.

This paper highlights the importance of higher education in the sustainability of development. The quantity of labor has a weakening impact on green GDP, while the quality of labor represented by education has exerted a more pronouncing influence on green economies. This implies that economic development starts to rely less on the number of labors in the workforce and become more dependent on higher education, which not only provides elements for production, but also effectively improves industrial structures and increases production efficiency.

There were two limitations of our studies. First, we have indirectly estimated green GDP by creating a new indicator, though the absolute value of this indicator cannot stand for the real green output. In spite of our meaningful exploration in the calculation of green GDP, further analysis and discussion about relevant indicators are still needed in order to establish the reliability and validity, considering the complexity in calculating environmental pollution and the unavailability of data about resource consumption when estimating green GDP. Second, we have proposed one possible interpretation of the influencing mechanism of higher education on green GDP based on current knowledge and speculation, and the effectiveness of this interpretation still needs to be evidenced by empirical studies in the future.

**Author Contributions:** Conceptualization and the research design, X.D., W.G., and R.C.; data curation, X.D. and R.C.; formal analysis, W.G. and R.C.; funding acquisition, X.D. and W.M.; investigation, W.G. and R.C.; resources, X.D. and W.M.; visualization, W.G. and R.C.; writing—original draft, W.G., X.D., and R.C.; writing—review and editing, X.D. and W.G.

**Funding:** This research was funded by a key research project (2016) of the Institute of Educational Economics, Peking University, "Education and Economic Growth under the Background of New Economic Normalization", sponsored by Ministry of Education, P.R.C., grant number: 16JJD880004.

**Acknowledgments:** The authors thank the support from the Institute of Educational Economics, Peking University.

**Conflicts of Interest:** The authors declare no conflict of interest.

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
