# Peer review of "An Empirical Study of the Role of Higher Education in Building a Green Economy"

_sustainability, doi:10.3390/su11236823_

Round 1
Reviewer 1 Report
This manuscript is very interesting and brings a new look on the relation between economic growth and sustainability. However the authors should improve the paper before acceptance, since there are many missing gaps that they must fit.
Authors organise the manuscript speaking in section 1 as the introduction where they insert the apparent research questions. After that they mixed up the meaning of sections with parts of theoretical background and methodologies used. This has no sense. Even it is strange that they introduce the research questions at the end of introduction and only on Section 3 propose the main hypotheses. And these: “Firstly, higher education has positive influence on building green economies; Secondly, higher education contributes more to green GDP compared with the tradition GDP.”
This second hypothesis doesn’t make sense, since GDP doesn’t contribute at all to green GDP. Or do you mean that “higher education contributes more to green GDP than to traditional GDP”?
2.1. has no reference at all. Although I understand this descriptive part is very important for defining chinese situation I would advise the authors to introduce some references to support those evidences/statements.
Solo model. Although people may know the meaning, you should in the first time you cited split the word solo for readers understand better.
You say “Since green GDP evolves from the traditional GDP, which is inextricably linked with higher education, there should also be a close relationship between green GDP and higher education”. Again, this assumption is not so clear as you want to pass. It also depends always on your definition of green GDP and this is something that it’s not yet standardised or even generalised in terms of concept.
You create a “new” indicator for green GDP that uses only energy. You forgot completely the resources since you based only that all resources are extracted with energy. That is ok, but where do you integrate the damage, pollution, land changes, biodiversity loss, interalia? So, my concern is how I can accept your definition or equation as green GDP. It is quite normal for you the assumption “Green GDP comes from and is tightly related to the traditional GDP”. If you consider only energy, of course it’s quite obvious.
Considering your figures as a summing up, in Figure 1 you consider that efficiency of energy is related with environmental preservation. Yes, it’s true but only in part. Again, you need to defend your model that everything function in base of energy. Which is true, but, you are not considering the whole landscape. In Figure 2 apparently you consider only primary industry not primary sector, or is it the same for you? If primary industry (in your figure) is decreasing who is making food for human growth?
Again there are missing aspects that you need to explore better.
Author Response
Thank you so much for your comments. please see the author response attached here.

Reviewer 2 Report
The appropriate identification of green economy drivers is one the key factors of its success. Therefore, I find the main purpose of the paper actual and very important.
The author analyzes in a thoughtful and well-structured way the role of higher education in building a green economy. After reading the paper, I fully agree with the statement that ‘this study provides empirical evidence for the substantial contribution that higher education makes in promoting green economic growth to achieve comprehensive sustainable development’.
I think, this paper should be published, with a minor changes:
1) Author should consider shortening the discussion section
2) In Discussion or Conclusions section author should recognize the limitations of their paper and emphasize problems for further research.
Author Response
Thank you so much for your comments. Please see the author response attached here.

Reviewer 3 Report
Comments for authors
The manuscript proposes a new approach to indirectly estimate green GDP, as well as, the impact of higher education on a green economy. I found merit in the detail and comprehensive study, but also had some minor concerns.
Some chapters are expanded, I would suggest the authors compact these part of manuscript. e.g. Chapter 2 (Line 76) and Chapter 3 (Line 164)
Could you add an explanation of the variables in the formulas (e.g. line 249-250).
I have no any particular comment to the methodology and results, however I have major comments about interlinking between Results and Discussion.
In Discussion, authors describe in detail mechanisms of higher education and situation in China. On the other side Results analysed all countries in the World and described the role of higher education on Green GDP. I miss detail discussion focused on results and I am not sure how results help to suggest mechanisms mentioned in the figure 1.
At the present form manuscript has described character sufficient scientific novelty and new approach. However, discussion is not sufficient interlinking. Therefore, I suggest a revision of the discussion part of manuscript.
Author Response

(The authors gave the same response as above.)

Round 2
Reviewer 1 Report
The authors have improved the paper but not enough to be immediately accepted. You need to be aware of your manuscript structures.
Why did you maintain the division of paper on sections if you add a new part 2 for literature review? line 57, 59, 69, 73 and 74.
However within your answer you said you have arranged it. Why then, maintain it? Although you tried to simplify this I don’t think you have been succeeded. When you add your research questions you should then introduced your work hypothesis. There is no sense to maintain a “section 3” as you said with 4 parts. The new indicator - as you said - is your output after all. And to test that output you use World data bank and apply that to a linear regression model Section 5 and 6 are your discussion and conclusions, doesn’t make sense the discrimination in part 1 (introduction).
In top of that (that are mine most important aspects that you need to revise) I can add some other particular comments that I didn’t mention before because I was mainly fixed in your manuscript arrangement. Regretfully, I should have mentioned that since my first revision.
What do you mean by GAP, although you have included a formula to justify the calculation? GAP stands for what? GDP is gross domestic production and GAP is ???
Looking at your Table 1 and trying to apply your formula for GAP = standardised green GDP - standardised GDP, I do not understand the values you got. For example - 0.86-(-0.03) = 0.89 and not -0.72, for Qatar and the same for others.
Also, you should give indication about the meaning of equation terms. For example, equation (5) and (9). It is also unclear why you use GAP in (9) and in Table 1 and 2 and not in Table 3.
Reviewer 3 Report
Questions raised in the manuscript are very relevant and fit well to the scope of the Sustainability journal. I think authors perform sophisticated analysis to support their conclusions.
I would like to thank authors to accept my suggestion and to explain some unclear issues.
Based on the above I accept the manuscript in this form and I recommend it for publishing.
Author Response
General Comments:
Questions raised in the manuscript are very relevant and fit well to the scope of the Sustainability journal. I think authors perform sophisticated analysis to support their conclusions.
I would like to thank authors to accept my suggestion and to explain some unclear issues.
Based on the above I accept the manuscript in this form and I recommend it for publishing.
General Response:
Thank you so much for your careful reading of our manuscript, and we highly appreciate your suggestions that have allowed us to greatly improve the quality of this paper.
With Best Regards,
Wenjuan Gao, Xiaohao Ding*, Ran Chen, Weifang Min
